# Biophysical and Structural Characterization of the Interaction between Human Galectin-3 and the Lipopolysaccharide from *Pseudomonas aeruginosa*

**DOI:** 10.3390/ijms25052895

**Published:** 2024-03-01

**Authors:** Luciano Pirone, Maria Pia Lenza, Sonia Di Gaetano, Domenica Capasso, Martina Filocaso, Rita Russo, Cristina Di Carluccio, Michele Saviano, Alba Silipo, Emilia Pedone

**Affiliations:** 1Institute of Biostructures and Bioimaging, National Research Council (CNR), Via P. Castellino 111, 80131 Naples, Italy; luciano.pirone@cnr.it (L.P.); sonia.digaetano@cnr.it (S.D.G.); martina.filocaso@unicampania.it (M.F.); rita.russo@unicampania.it (R.R.); 2Department of Chemical Sciences, University of Naples Federico II, Via Cinthia 4, 80126 Naples, Italy; mariapia.lenza@unina.it (M.P.L.); cristina.dicarluccio@unina.it (C.D.C.); 3Interuniversity Research Centre on Bioactive Peptides (CIRPEB), University of Naples Federico II, 80134 Naples, Italy; domenica.capasso@unina.it (D.C.); michele.saviano@cnr.it (M.S.); 4Department of Physics “Ettore Pancini”, University of Naples Federico II, Via Cinthia 4, 80126 Naples, Italy; 5Department of Environmental, Biological and Pharmaceutical Sciences and Technologies, University of Campania “Luigi Vanvitelli”, 81100 Caserta, Italy; 6Institute of Crystallography, National Research Council (CNR), 81100 Caserta, Italy

**Keywords:** galectin-3, LPS, interaction studies, NMR, ITC

## Abstract

Given the significant involvement of galectins in the development of numerous diseases, the aim of the following work is to further study the interaction between galectin-3 (Gal3) and the LPS from *Pseudomonas aeruginosa*. This manuscript focused on the study of the interaction of the carbohydrate recognition domain of Gal3 with the LPS from *Pseudomonas aeruginosa* by means of different complementary methodologies, such as circular dichroism; spectrofluorimetry; dynamic and static light scattering and evaluation of the impact of Gal3 on the redox potential membranes of *Escherichia coli* and *P. aeruginosa* cells, as well as ITC and NMR studies. This thorough investigation reinforces the hypothesis of an interaction between Gal3 and LPS, unraveling the structural details and providing valuable insights into the formation of these intricate molecular complexes. Taken together, these achievements could potentially prompt the design of therapeutic drugs useful for the development of agonists and/or antagonists for LPS receptors such as galectins as adjunctive therapy for *P. aeruginosa*.

## 1. Introduction

Virtually all bacterial and eukaryotic cells, as well as many viruses, display surface glycans, which act as regulators of a variety of biological events and mediate host–microbe interactions, including immunomodulation and inflammation processes, through their recognition by specific glycan-binding proteins, mainly known as lectins [1]. Microbial lectins are involved in host colonization, whereas some animal lectins can mediate immune recognition of microbial and parasite envelope glycans and promote events like activation and regulation, mediating immunomodulation and inflammation processes. Within the lectin family, galectins represent an evolutionarily conserved group of proteins with the ability to bind β-galactosides via characteristic carbohydrate recognition domains (CRDs) [2,3,4,5,6].

These proteins play an important role in several biological processes [7] and are therefore becoming emerging targets for diagnostic and therapeutic approaches, and several inhibitors, mainly of a glycosidic nature, have been identified and characterised [8,9,10,11,12,13].

Galectins can act as pathogen recognition receptors against a wide range of microorganisms, interacting directly with bacterial surface glycans and mediating the recognition and effector functions in innate immunity [14,15]. Both Gram-positive bacteria, such as *Streptococcus pneumoniae*, and Gram-negative bacteria, such as *Klebsiella pneumoniae* and *Pseudomonas aeruginosa*, display surface carbohydrate galectin ligands [1,16]. Galectin-3 (Gal3) is structurally unique among all galectins, as it contains a C-terminal CRD linked to an N-terminal protein-binding domain, being the only chimeric galectin [17,18,19]. In particular, although it is widely expressed in human tissues, its functions seem to strictly depend on its subcellular compartmentalization [20,21,22,23]. In detail, it is demonstrated that extracellular Gal3 mediates cell adhesion and cell–cell interaction through specific recognition of complex carbohydrates on the cell surface, while intracellular Gal3 is implicated in cell apoptosis, autophagy and inflammation [24,25,26,27]. Interestingly, recent studies suggest that Gal3 is involved in cell metabolism and linked to diabetes and cancer [28,29,30]. Indeed, it has been reported that Gal3 deficiency is associated with the dysregulation of glucose metabolism and leads to hyperglycemia; therefore, it is proposed that Gal3 benefits glucose homeostasis and has a protective effect on diabetogenesis when nutrients are in excess.

Lipopolysaccharide (LPS) [31,32,33], the main component of the outer membrane of Gram-negative bacteria, is recognized by the immune system as a marker of bacterial invasion [16,20,24,25,34]. LPS immune recognition stimulates the production of inflammatory cytokines, thus activating the immune response. In addition, the blood levels of LPS fluctuate with the gut microbiota, and an elevated LPS level is associated with subclinical chronic inflammatory processes, obesity, impaired glucose metabolism and even cancer [21,35]. How LPS influences glucose metabolism and is linked to diabetes and cancer has been studied by Chen X., et al. (2022) [20], who reported that intracellular Gal3 senses LPS to lead to the activation of mTORC1 signaling [20]. It has also been reported that LPS interacts with Gal3 to regulate the non-canonical inflammasome [36,37]. This is consistent not only with the data from the literature suggesting that Gal3 is a sensor of LPS but also with observations that the LPS/Gal3 interaction is involved in the development of diabetes and cancer, with both disease states closely associated with inflammatory responses.

In this context, it is worth mentioning that previous studies have indicated that Gal3 binds LPSs of several bacterial species, including *P. aeruginosa* [38,39]. In the lungs of cystic fibrosis (CF) patients, chronic infection by *P. aeruginosa* induces excessive inflammation, which not only damages the lungs but also contributes to an inability to eradicate infection. Although it is well established that Gal3 can interact with LPS, the pathophysiological importance of the LPS/Gal3 interaction is not fully understood; therefore, here, we report a detailed characterization of the interaction between the CRD of Gal3 (Gal3^CRD^) and the LPS from *P. aeruginosa* 10 (LPS*pa*) using different biophysical techniques with the aim of a better comprehension of the interaction between them.

## 2. Results

### 2.1. Spectroscopic Analyses

To analyze the interaction between Gal3^CRD^ and LPS from *P. aeruginosa*, 10 circular dichroism and spectrofluorometric analyses were carried out (Figure 1).

As already described [9], the spectrum of Gal3 is not a typical spectrum of a protein with beta-sheets but displays particular characteristics in its topological arrangement, such as the length of the filaments, intra/intersheet twists and β-turns producing a spectrum with a minimum of around 220 nm. The far-UV Gal3^CRD^ spectrum registered in the presence of an increasing concentration of LPS*pa* showed the partial denaturation of Gal3^CRD^, as corroborated by the disappearance of the positivity at around 200 nm.

The titration experiments showed a decrease in fluorescence emission as a function of LPS*pa* concentration, demonstrating that interaction with LPS*pa* truly takes place and suggesting the occurrence of an LPS*pa*-induced conformational change toward a more compact structure. Moreover, since no λ shift can be observed in the fluorescence spectra upon LPS*pa* binding, it is possible to argue that major modifications in protein hydrophobicity are not required to transduce the structural modifications.

### 2.2. DLS and ITC Studies

To estimate the average size of the LPS*pa* particles, the hydrodynamic radius (rh) was measured using the dynamic light scattering technique (DLS).

DLS is a well-known technique used to measure Brownian motion (diffusion) and the size distribution of particles in solution. For this reason, DLS experiments were used to investigate whether Gal3^CRD^ interacts with LPS*pa* (above their critical micelle concentration, i.e., 0.6 μM.) LPS*pa* alone was initially present according to a size distribution with an average diameter of around 20 nm. The variation in size of Gal3^CRD^ with LPS*pa* at a ratio of 1:10 was investigated until the peak corresponding to Gal3^CRD^ alone disappeared, which could be explained by the formation of a complex between the protein and LPS*pa* (Figure 2).

In addition, in order to investigate how Gal3^CRD^ interacts with LPS*pa* and whether the protein can alter the size of the micelles of LPS*pa*, the aggregation behavior of LPS*pa* in the presence of an increasing concentration of Gal3^CRD^ was analyzed. A reduction in the LPS*pa* diameter from 11.2 nm (LPS*pa* 10 μM alone) to 7.1 nm in the presence of Gal3^CRD^ (LPS*pa*—Gal3^CRD^ molar ratio of 1:3) was observed, suggesting the disaggregating effect of the protein (Figure 3).

The disaggregating effect of Gal3 on LPS*pa* could significantly lower its biological activity. The data from the literature [40] show that the biological activity of antimicrobial peptides lies in their potent activity in detoxifying LPS through the breakdown of LPS aggregates. The activity of Gal3 against LPS can be hypothesized in this context.

Considering that the bacterial cell surface is the first variable typically defined in studying bacteria–molecule binding, zeta potential (ZP) measurements can be used as a reporter for such interactions. Therefore, ZP studies were carried out to monitor the effect of Gal3^CRD^ on the membrane surface charge of the *E. coli* and *P. aeruginosa* cells. The *E. coli* cells displayed a zeta potential of about −15.2 mV. Upon the addition of increasing concentrations of Gal3^CRD^ until 35 μM, the *E. coli* ZP values increased and then stabilized at approximately −5.6 (Figure 4). The same trend was observed in the presence of the *P. aeruginosa* cells, whose ZP increased from −10.3 mV to −5.6 in the presence of Gal3^CRD^. Therefore, in both cases, we observed an increase in the ZP toward neutral values due to the interaction between Gal3^CRD^ and both LPSs.

Finally, a more detailed characterization of the interaction of Gal3^CRD^ with LPS*pa* was carried out using isothermal titration calorimetry (ITC) (Figure 5). The ITC experiment showed a sequential binding site, revealing a Kd of a low μM value. The best fit is obtained by adding three sequential binding molecules. The first interaction is clearly the most affine, showing a K_D_ = 6.0 ± 0.5 μM (Table 1).

### 2.3. NMR Studies

To elucidate the intricate dynamics of the interaction between LPS and Gal3, we performed additional Nuclear Magnetic Resonance (NMR) experiments, employing ^15^N-^1^H TROSY experiments. This approach enabled us to probe the behavior of Gal3^CRD^ in the absence of and upon the addition of the *P. aeruginosa* LPS. Notably, ligand binding induced modifications in the chemical shifts of the protein’s amide signals, prompting the tracking of these chemical shift perturbations for deeper insights into the structural aspects of molecular recognition events.

Upon adding 0.2 equivalents of LPS*pa* to ^15^N-Gal3^CRD^, a general decrease in the peak intensities was observed (see Figure 6). This behavior can be attributed to the increase in the relaxation time of Gal3^CRD^ upon binding, which led to line broadening and a consequent loss of signal intensity.

In Figure 7, a plot illustrating the variation in amino acid intensity after the addition of LPS*pa* was shown. Several affected amino acids were in the canonical binding site on the S-face, within the β-sheets S4–S5. Remarkably, high perturbations were noticed in residues crucial to lactose binding, such as His51 and Trp74, which disappeared upon LPS*pa* binding. This observation suggested that these residues played a pivotal role in LPS binding. Furthermore, β-sheets S2 and S6 exhibited perturbations, indicating that the epitope-binding mode was extended, as expected. Furthermore, other sheets on the F-face of the CRD displayed significant perturbations upon binding. Particularly, polar amino acids were highly sensitive to the binding process.

The reduction in peak intensity could be explained by the establishment of supermolecular lectin–LPS complexes driven by multivalent presentations. Indeed, although the interaction was mediated by the canonical binding site, the formation of these supermolecular complexes impacted the molecular tumbling of the proteins, resulting in decreased and broadened signals.

Based on the experimental results (Figure 7) and obtaining further insights from the 3D structure of Gal3^CRD^ (Figure 8), it is evident that the primary binding site with LPS*pa* is located on the S-face of the CRD, involving key residues (His51, Trp74, Asn57, Asn72, Asn73). However, the amino acids situated at the top of the protein also exhibit line broadening, and, specifically, the most perturbed amino acids are charged (Arg61, Asp71, Glu77, Lys89, Asp132 and Arg117) and apolar residues (Asn107, Leu121, Asn122 and Ser137), along with hydrophobic residues (Phe85).

## 3. Discussion

The rise of multidrug-resistant bacteria and the formation of biofilms that evade the host immune response, leading to an increasing number of hospital infections, represent major health concerns [38]. The lack of new antibiotics, particularly those that have different mechanisms of action and that are active against Gram-negative bacteria, has exacerbated the situation. In this context, the search for drugs that act on new targets is a crucial challenge. The available drugs, mainly of a peptidic nature, are known to act on the bacterial membranes and/or lipopolysaccharide (LPS) of *P. aeruginosa*, a Gram-negative bacterium frequently associated with severe infections in immunocompromised hosts or in patients with cystic fibrosis. However, their use in clinical practice is presently limited because of their toxicity, the cost of their synthesis and, for some of them, their susceptibility to proteolysis [41].

Here, we report that Gal3 is able to interact with LPS and induce membrane depolarization in *P. aeruginosa*, so the LPS/Gal3 complex could be considered a new target for drugs that can inhibit LPS activity. Although the data from the literature report the interaction between Gal3 and LPS [39], new experiments have been designed to shed more light on the mode of interaction and the sites directly involved in the binding process. Here, experiments with DLS, ITC and NMR spectroscopy were conducted. The data obtained from the DLS measurements confirmed the presence of an interaction because of the disappearance of the peak relative to the hydrodynamic radius of Gal3^CRD^ alone when complexed with LPS*pa*. In addition, it is worth mentioning that Gal3 seems to exert a disaggregating effect on LPS. Furthermore, a further experiment regarding the determination of the membrane zeta potential of *E. coli* and *P. aeruginosa* cells in the absence and presence of Gal3^CRD^ confirms that the interaction between the two molecules results in the hyperpolarization of the membrane by Gal3^CRD^.

An isothermal titration calorimetry analysis showed a sequential mode of binding sites in which the first, most affine interaction showed an affinity constant value of 6 μM; these data may represent a good experimental approach to selecting Gal3 inhibitory molecules with a potential anti-inflammatory effect. Finally, the hypothesis of binding between Gal3^CRD^ and LPS gained support from the NMR data analysis. The analysis of the protein perspective reveals the involvement of Gal3’s S-face. Specifically, the key residues essential to binding with carbohydrates within the S3–S4–S5 β-sheet exhibit significant perturbations. Notably, residues in S2 and S6 are also affected, suggesting an extended binding interaction mode. Moreover, the interaction seems to be facilitated by the polar amino acids situated on the upper surface of Gal3^CRD^. Therefore, we can hypothesize that, besides the canonical binding site (S3–S4–S5), LP*Spa* also establishes further interactions atop of Gal3^CRD^ (Figure 8). This comprehensive analysis supports the hypothesis of an interaction between Gal3 and LPS, unraveling the structural details and providing valuable insights into the formation of these intricate molecular complexes.

In this paper, we have elucidated the intricate interplay between the C-terminal domain of Gal3 and the LPS from *P. aeruginosa*. Notably, the presence of the N-terminal domain promotes the oligomerization of Gal3, enhancing neutrophil activation [37]. Consequently, the interaction of the LPS and full-length Gal3, in its oligomeric state, could improve the binding of the LPS to the cell surface, decreasing the activation threshold of the neutrophils in response to LPS. In this regard, as for intracellular bacteria, the C-terminal domain of Gal3 is essential to increasing the LPS-induced assembly of the intracellular caspase-4/11 oligomers and to their activation. Conversely, the N-terminal domain contributes to the self-association property of Gal3 and amplifies the intracellular immune responses of caspases, which rely on the functional multivalency of Gal3.

Our observations, based on the experimental data, suggest that the mechanism of action is highly dependent on the ratio of Gal3^CRD^ to LPS, which could be important in the context of bacterial infections. Altogether, these results could lead to the design of therapeutic drugs useful in the development of agonists and/or antagonists for LPS receptors such as galectins as adjunctive therapy for *P. aeruginosa*. It is evident that the data collected concern the LPS from *P. aeruginosa* and do not exclude the proposition that the same experiments, conducted on LPSs from different sources, may yield different and/or conflicting results. This topic needs to be thoroughly investigated and will represent an interesting field of future research.

## 4. Materials and Methods

### 4.1. Protein Expression

The human galectin-3 CRD (named Gal3^CRD^) used in this study was produced in *E. coli*, as previously described [8]. To express unlabeled protein, the bacteria growth was carried out in LB medium. Protein expression and purification protocols were implemented for labeled Gal3^CRD^ production. In detail, M9 minimal medium supported with 0.5 g/L of ^15^NH_4_Cl is prepared for the expression of the ^15^N-labeled protein. A colony of the *E. coli* strain BL21(DE3) GOLD transformed with the recombinant vector pETM11/Gal3^CRD^ is taken from the plate and inoculated into 100 mL of LB. After over-night growth, the pre-inoculum is centrifuged at 6800 rpm for 10 min at 4 °C, the LB is removed and the pellet is resuspended with 10 mL of M9 medium. It is transferred into 1 L of pre-warmed M9 medium and incubated at 37 °C until induced to an optical density of 0.6 to 0.8 OD600 nm using a final IPTG concentration of 1 mM. Induction is followed by incubation at 25 °C for 16–18 h. The subsequent steps follow the protocol elsewhere described [42]. The protein sample is brought to a concentration of 200 μM in 50 mM Tris-HCl, 150 mM NaCl and 1 mM DTT pH 7.5.

### 4.2. LPS Preparation

LPS is a heterogeneous molecule and tends to form aggregates of varying sizes. However, when treated with detergents, ultrasound and heat, a population of molecules with molecular weights between 30 kDa and 100 kDa can be obtained. In our experiment, 1 mg of LPS from *P. aeruginosa* 10 (L9143, SIGMA-Aldrich, St. Louis, MO, USA) was resuspended in H_2_O stearyl or in 20 mM sodium phosphate and 150 mM NaCl at a pH 7.4. The solution was mixed with the aid of a vortex mixer and then sonicated at 50 °C for 30 min in the sonicator bath. After treatment, the LPS MW was checked using SDS-PAGE with silver nitrate gel staining. In our experiments, solutions from 0.5 to 20 mM were used.

### 4.3. Spectroscopic Analyses

The CD spectra were measured using a Jasco J-1500 spectropolarimeter equipped with a Peltier thermostatic cell holder (Jasco Europe, Cremella, LC, Italy). The measurements were performed at 20 °C using a 0.1 cm path length cell in 10 mM sodium phosphate and 1 mM DTT at a pH of 7.4. The far-UV CD spectra were monitored from 195 to 260 nm using Gal3^CRD^ final concentrations of 10 μM. The far UV spectrum was registered in the presence of an increasing concentration of LPS*pa* (0–10 μM). The CD spectra were averaged over at least three independent scans and the baselines corrected by subtracting the buffer contribution. The spectrofluorometric Gal3^CRD^ spectra at a concentration of 10 μM were registered using a Jasco FP-750 (Jasco Europe, Cremella, LC, Italy). The sample was excited at 280 nm and emissions registered between 300 and 400 nm. The spectrofluorimetric spectra were registered in the presence of an increasing concentration of LPS*pa* (0–10 μM).

### 4.4. Dynamic Light Scattering (DLS) Analyses

The DLS measurements were carried out using a Malvern nanozetasizer (Malvern, UK). The samples were placed in a disposable cuvette and held at 37 °C. The Gal3^CRD^ was assayed at a concentration of 200 μM, while the LPS*pa* was studied at 10 and 20 μM. In the interaction studies, a Gal3^CRD^/LPS*pa* ratio of 10:1 was analyzed, while in the aggregation studies, LPS*pa*/Gal3^CRD^ ratios of 1:1 and 1:3 were investigated. For each sample, the analyses were recorded three times with 11 sub-runs using the multimodal mode. The Z-average diameter was calculated from the correlation function using the Malvern technology software ZS Xplorer version 3.2.0.84.

### 4.5. Zeta Potential Measurements

The *E. coli* and *P. aeruginosa* cells in the mid-logarithmic phase were diluted to an OD600 nm of 0.005 (50,000 cells). The volume of the cells (700 μL) was harvested and measured. Then, increasing concentrations of Gal3^CRD^ from 0 to 35 μM were added to the cells, and the potential was measured. The samples were placed in cuvettes equipped with instrument-specific gold electrodes. For each concentration, a total of 3 measurements of 100 runs each were carried out. The experiments were carried out on the zetasizer Nano ZS (Malvern Instruments, Malvern, Worcestershire, UK) equipped with a 633 nm He laser. The statistical significance was determined using Student’s t-test (paired, two-sided), and a p value less than 0.05 was considered to be significant.

### 4.6. Isothermal Titration Calorimetry (ITC)

The ITC experiments were conducted at a temperature of 37 °C using a MicroCal PEAG-ITC (Malven Panalytical, Malvern, UK). Titration was conducted using Gal3^CRD^ as the ligand at a concentration of 275 µM and titrating the LPS*pa* (in cell) at a concentration of 20 µM. The LPS*pa* and Gal3^CRD^ were prepared in the same buffer (20 mM sodium phosphate, 150 mM NaCl, 1 mM DTT, pH 7.4). The titration was designed so that an injection occurred every 150 s, for a total of 27 injections of 1.5 μL (except for the first injection of 0.4 uL), at a stirring speed of 1000 rpm. Finally, to exclude the presence of non-specific heat, two control titrations were performed: Gal3^CRD^ was injected into the cell containing only buffer; buffer was injected into the cell containing LPS*pa*. The data were reprocessed using ITC Data Analysis in the Origin software version 7.0 and imposing a sequential binding site model. The best data fitting was obtained by adding three sequential binding molecules.

### 4.7. NMR

The NMR experiments were recorded using a Bruker AVANCE NEO 600 MHz equipped with a cryo probe (Bruker Italia Srl, Milano, Italy), and the data acquisition and processing were performed using TopSpin software v. 4.1.1. All the NMR experiments were conducted at 25 °C. The samples were dissolved in 500 μL of 20 mM of Tris buffer, 150 mM of NaCl and 1 mM of DTT at a pH of 8 using 5 mm NMR tubes. Spectra with 75 µM of uniformly ^15^N-labeled Gal3^CRD^ were recorded in the apo form and after the addition of LPS*pa* (5:1 Gal3^CRD^: LPS*pa*). A TROSY experiment was used, in which 32 scans were acquired with 256 (t1) × 2048 (t2) complex data points in the ^15^N and ^1^H spectra, respectively. The CcpNmr Analysis software v. 3.2.0 was employed for the data analysis [43,44]. The average intensity changes were calculated using the following equation: % perturbation = I_i_ − I_f_/Δ_max_.

## Figures and Tables

**Figure 1 ijms-25-02895-f001:**
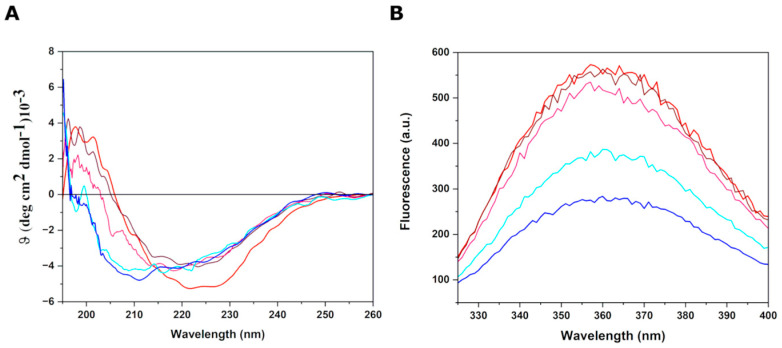
(**A**) Circular dichroism measurements. Overlay of far-UV CD spectra of Gal3^CRD^ alone (in red) and in presence of increasing concentration of LPS*pa* was reported (0.5–10 μM). (**B**) Fluorescence emission analysis. Overlay of Gal3^CRD^ spectra alone (red line) and in presence of increasing concentration of LPS*pa* (0.5–10 μM) was shown: 0.5 μM (brown line); 2.5 μM (purple line); 5 μM (cyan line); 10 μM (blue line).

**Figure 2 ijms-25-02895-f002:**
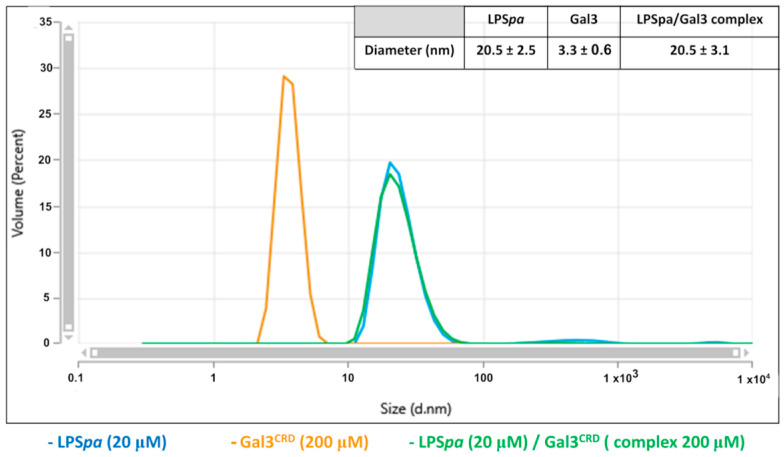
DLS measurements. DLS measurements were carried out using a Zetasizer Nano ZS (Malvern Instruments, Westborough, MA, USA) equipped with a 173° backscatter detector, at 37 °C, using a disposable sizing cuvette. Data were analyzed using the software OmniSIZE (Viscotek) 2.0. DLS measurements in triplicate were carried out on aqueous LPSpa samples at 20 μM. LPS*pa* size measurements were performed before and after Gal3^CRD^ addition (200 μM).

**Figure 3 ijms-25-02895-f003:**
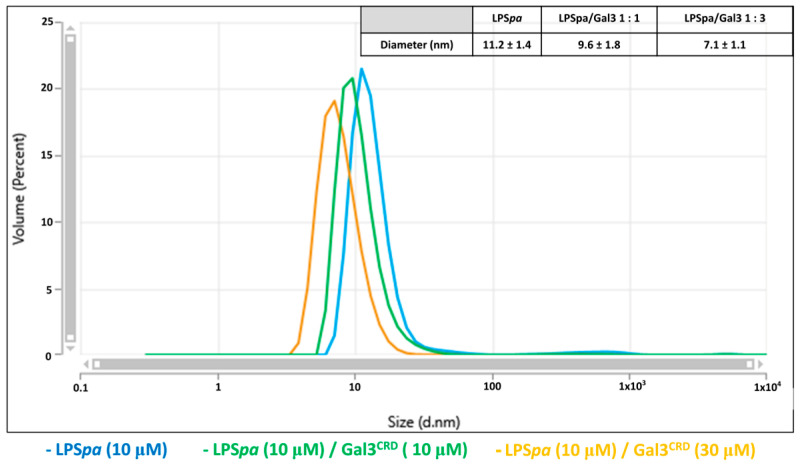
DLS measurements. LPS*pa* (10 μM) size measurements were performed alone (blue line) and in presence of Gal3^CRD^ (molar ratio 1:1, green line; molar ratio 1:3, orange line).

**Figure 4 ijms-25-02895-f004:**
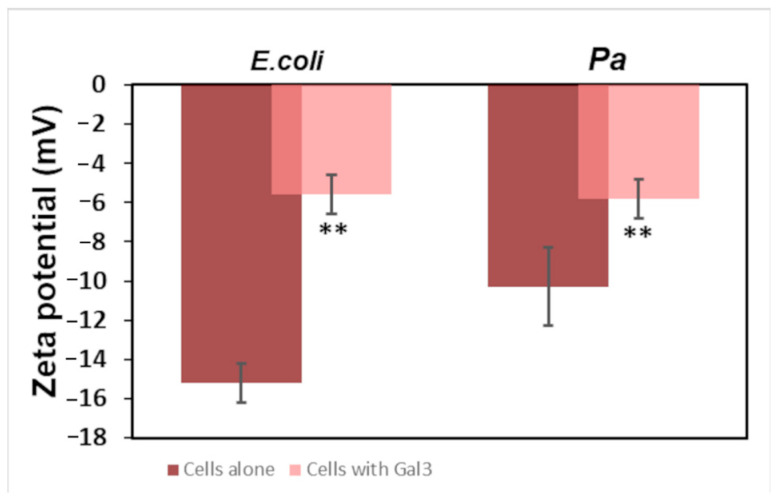
Zeta potential analysis. The potential values of *E. coli* and *P. aeruginosa* cells are shown alone (brown bar) and in the presence of Gal3^CRD^ 35 μM (pink bar). ** *p* < 0.01.

**Figure 5 ijms-25-02895-f005:**
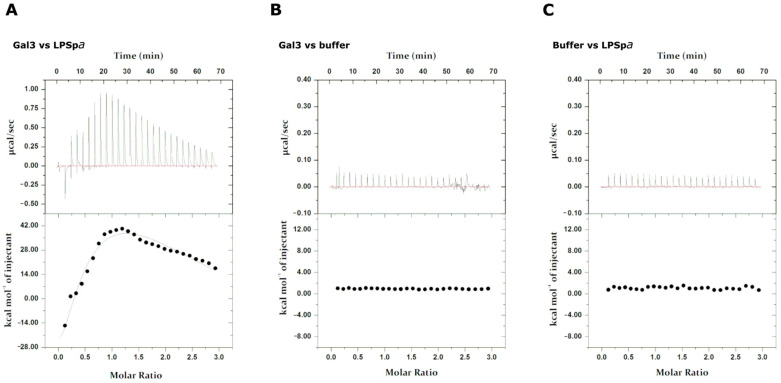
Isothermal titration calorimetry analysis. In (**A**), titration of LPSpa with Gal3^CRD^; (**B**,**C**): the negative controls (titration of buffer with Gal3^CRD^) in (**B**) and titration of LPS*pa* with buffer in (**C**) are shown. The top and bottom panels report raw and integrated data, respectively.

**Figure 6 ijms-25-02895-f006:**
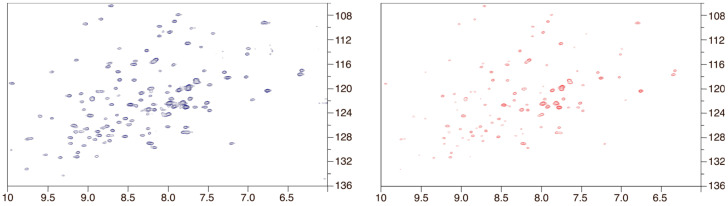
^1^H-^15^N TROSY spectra are shown for ^15^N-Gal3^CRD^ alone (cross-peaks in blue) and in the presence of 0.2 eq of LPS*pa* (cross-peaks in red).

**Figure 7 ijms-25-02895-f007:**
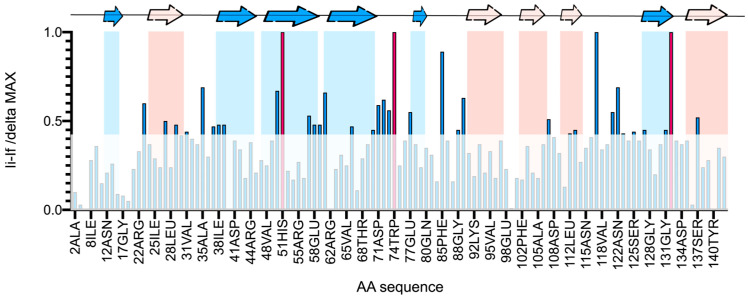
TROSY resonance broadening maps showing the binding interactions between Gal3^CRD^ and LPS*pa*. Changes in Gal3^CRD^ resonance intensities observed for Gal3^CRD^ in the presence of LPS*pa* are plotted vs. the amino acid sequence of Gal3^CRD^. Resonance intensity changes are quantified as fractional adjustments by subtracting the intensity of a specific TROSY cross-peak in the apo form spectrum from that in complex with LPSpa. The resulting difference is then divided by the maximal differences observed. Any amino acid that disappears after binding is colored in red.

**Figure 8 ijms-25-02895-f008:**
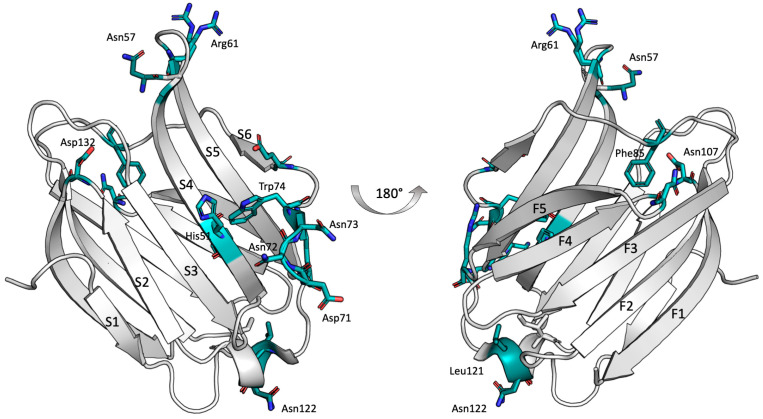
The primary binding surface on the CRD of Gal3 is shown. Segments containing residues that are most affected by binding to LPS*pa* are highlighted in light blue on the structure of Gal3 (pdb access code: 3ZSJ).

**Table 1 ijms-25-02895-t001:** Thermodynamic parameters of the titration of LPS*pa* with Gal3^CRD^.

Ligand	K_D1_	ΔH_1_	ΔS_1_
Gal3^CRD^	6.0 ± 0.5 μM	−3.2 ± 0.1 kcal/mol	−0.1 kcal/mol/deg

## Data Availability

Data contained within the article.

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
