# Peer review of "Biophysical and Structural Characterization of the Interaction between Human Galectin-3 and the Lipopolysaccharide from Pseudomonas aeruginosa"

_ijms, 2024, doi:10.3390/ijms25052895_

Round 1

Reviewer 1 Report

Comments and Suggestions for Authors

Reviewer comments:

In this manuscript the authors have investigated the interaction between carbohydrate recognition domain of Galectin-3 and LPS from Pseudomonas aeruginosa. Interactions between these molecules are investigated using different methodologies such as circular dichroism, spectrofluorimetry, dynamic and static light scattering, ITC and NMR studies. The introduction provides a good background , but results and discussion sections should be corrected to be more clear for the readers. Here are a few suggestions that will improve the quality of the manuscript:

Please, change the number in the superscript after the authors` names.

In the abstract section insert abbreviations after Galectin-3. Use the same abbreviation (Gal3 or Gal-3) for Galectin-3 in a whole manuscript.

Line 76

Please, insert the references after cancer.

Line 78

Check the number of reference (20 or 35)

Line 89

Delete Carbohydrate Recognition Domain

Line 101-105

The changes of the far UV CD spectra of GalCRD with increasing concentration of LPS is missing. Please replace the figure 1A with the new one in which the transition of Gal3 structure is visible with increasing concentration of LPS .

Also in Figure 1B add spectra with an increasing concentration of LPS.

In section 2.2., specify what was the critical micelle concentration of LPS. According to the results, both concentrations of LPS  used in the experiment were above the critical micelle concentration, and that significantly changed the size of LPS micelles without Gal3CRD. Also in this section for Figure 2, was the final concentration of Gal3CRD in the mixture 200μM? How did you reach this concentration of Gal3CRD in a mixture, when in line 272 you stated that the concentration was 75 μM?

Line 150

Please in the manuscript use full names or abbreviations Escherichia coli and Pseudomonas 150 aeruginosa cells. E. coli cells

Line 154 

Remove mV after -5.6.

Section 2.3. NMR Studies

Please,  connect results from figure 7 and 8. Insert three letters abbreviations for amino-acids in the text and also add marked amino acids on the figure 8.

Could you expand the explanation of the contribution of Gly131 and S2  in binding Gal3 to LPS in the results section? 

Please in the discussion section add the possible influence of interaction between LPS and Gal3  when it is in solution compared to the same protein in the membrane.

Line 220 

GM, what is that?

Line 264

15 in superscript before NH4Cl, Use IU system for units (L instead of l).

Line 276 

Please insert concentration and pH of the buffer, and volume.

Line 280, 287, 317, 318, 324

Please insert the company and country of instrument .

Line 289

Insert DLS after Dynamic Light scattering

Line 291, 319

Convert K into °C.

Line 299

The volume of cells (700 μL) were….

Line 300

Insert a range of concentrations from XX to 35…

Line 306

Replace Isothermal calorimetry titration with ICT

Author Response

Dear Editor,

please find enclosed the revised version of our manuscript entitled “Biophysical and structural characterization of the interaction between human Galectin-3 and LPS from Pseudomonas aeruginosa” to be considered for publication in the journal “International Journal of Molecular Sciences”.

We thank the referees for the insightful comments and have done everything possible to accommodate their requests and improve the manuscript. You can find below point by point all the response to reviewers.

The main changes in the manuscript are highlighted in red in the word file

I referee

Reviewer comments:

In this manuscript the authors have investigated the interaction between carbohydrate recognition domain of Galectin-3 and LPS from Pseudomonas aeruginosa. Interactions between these molecules are investigated using different methodologies such as circular dichroism, spectrofluorimetry, dynamic and static light scattering, ITC and NMR studies. The introduction provides a good background , but results and discussion sections should be corrected to be more clear for the readers. Here are a few suggestions that will improve the quality of the manuscript:

Please, change the number in the superscript after the authors` names.

Done

In the abstract section insert abbreviations after Galectin-3. Use the same abbreviation (Gal3 or Gal-3) for Galectin-3 in a whole manuscript.

Done

Line 76

Please, insert the references after cancer.

Inserted references

Line 78

Check the number of reference (20 or 35)

Deleted Reference 35 and inserted ref 20

Line 89

Delete Carbohydrate Recognition Domain

Done

Line 101-105

The changes of the far UV CD spectra of GalCRD with increasing concentration of LPS is missing. Please replace the figure 1A with the new one in which the transition of Gal3 structure is visible with increasing concentration of LPS .

Also in Figure 1B add spectra with an increasing concentration of LPS.

Both figures  and corresponding legends were modified (Figure 1A and B) following referee suggestion.

In section 2.2., specify what was the critical micelle concentration of LPS. According to the results, both concentrations of LPS  used in the experiment were above the critical micelle concentration, and that significantly changed the size of LPS micelles without Gal3CRD. Also in this section for Figure 2, was the final concentration of Gal3CRD in the mixture 200μM? How did you reach this concentration of Gal3CRD in a mixture, when in line 272 you stated that the concentration was 75 μM?

We thank the reviewer and we have corrected the sentence in line 272 accordingly.

Line 150

Please in the manuscript use full names or abbreviations Escherichia coli and Pseudomonas 150 aeruginosa cells. E. coli cells

We have corrected the abbrevations

Line 154 

Remove mV after -5.6.

Done

Section 2.3. NMR Studies

Please,  connect results from figure 7 and 8. Insert three letters abbreviations for amino-acids in the text and also add marked amino acids on the figure 8.

Done

Could you expand the explanation of the contribution of Gly131 and S2  in binding Gal3 to LPS in the results section? 

During the titration of Gal3 with LPS, Gly131 was not perturbed; the aminoacid Asp132 was instead subjected to the CSP as indicated in Fig. 7 and highlighted in the text.

Please in the discussion section add the possible influence of interaction between LPS and Gal3  when it is in solution compared to the same protein in the membrane.

Done

Line 220 

GM, what is that?

Deleted

Line 264

15 in superscript before NH4Cl, Use IU system for units (L instead of l).

Done

Line 276 

Please insert concentration and pH of the buffer, and volume.

Done

Line 280, 287, 317, 318, 324

Please insert the company and country of instrument .

Done

Line 289

Insert DLS after Dynamic Light scattering

Done

Line 291, 319

Convert K into °C.

Done

Line 299

The volume of cells (700 μL) were….

Corrected

Line 300

Insert a range of concentrations from XX to 35…

Inserted

Line 306

Replace Isothermal calorimetry titration with ICT.

Sorry for the mistake, We have replaced as follows: Isothermal titration calorimetry (ITC)

Reviewer 2 Report

Comments and Suggestions for Authors

The authors present a study of the interaction of the carbohydrate recognition domain from galectin-3 with LPS from P. Aeruginosa using a set of biophysical methods including circular dichroism, fluorescence, DLS, zeta potential, ITC and NMR. The goals of the study are important and the application of a broad set of methods is admirable. The aspect of this paper that must be improved is the very brief and not too informative presentation of some of the results.

(1) CD measurements - Figure 2 is 'in the presence of LPSpa', how much? what is the interpretation of the 'de-structuration'? This means partial denaturation?

(2) Figure 2B, it says this is a a titration experiment but only two curves are shown and why the fluorescence decreases is not discussed.

(3) A CMC fpr LPS is referred to on page 4. What is cmc? The discussion of figure 3 needs to be made more complete

(4) line 164, please discuss sequential binding site model for the interaction in more detail

Comments on the Quality of English Language

The language is ok but please proofread.

Author Response

Dear Editor,

please find enclosed the revised version of our manuscript entitled “Biophysical and structural characterization of the interaction between human Galectin-3 and LPS from Pseudomonas aeruginosa” to be considered for publication in the journal “International Journal of Molecular Sciences”.

We thank the referees for the insightful comments and have done everything possible to accommodate their requests and improve the manuscript. You can find below point by point all the response to reviewers.

The main changes in the manuscript are highlighted in red in the word file

II referee

The authors present a study of the interaction of the carbohydrate recognition domain from galectin-3 with LPS from P. Aeruginosa using a set of biophysical methods including circular dichroism, fluorescence, DLS, zeta potential, ITC and NMR. The goals of the study are important and the application of a broad set of methods is admirable. The aspect of this paper that must be improved is the very brief and not too informative presentation of some of the results.

(1) CD measurements - Figure 2 is 'in the presence of LPSpa', how much? what is the interpretation of the 'de-structuration'? This means partial denaturation?

(2) Figure 2B, it says this is a a titration experiment but only two curves are shown and why the fluorescence decreases is not discussed.

Regarding point 1 and 2, we perfectly agree with referee that it was not enough clear so we have modified the figures adding the spectra in different concentration of LPS as suggested. In addition we have substituted the term de-structuration with partial denaturation.

(3) A CMC fpr LPS is referred to on page 4. What is cmc? The discussion of figure 3 needs to be made more complete

CMC means critical micelle concentration and for LPS corresponds to 0.66 mM. Such information has been added in the text. In addition, a sentence has been added to comment the result.

(4) line 164, please discuss sequential binding site model for the interaction in more detail.

A sentence has been added in the Results and Materials and Method section.

Reviewer 3 Report

Comments and Suggestions for Authors

The manuscript (ijms-2856899) examined the interaction between the carbohydrate recognition domain (CRD) of human galectin-3 and LPS from Pseudomonas aeruginosa by using different methods including circular dichroism (CD), fluorescence, dynamic light scattering (DLS), zeta potential measurements, isothermal titration calorimetry (ITC) and 15N-1H TROSY NMR spectroscopy. It was concluded that these findings could help drug design targeting galectin-LPS interaction for therapy against P. aeruginosa.

Overall, my major issue with the manuscript is on data quantification. Most of the data are only described qualitatively without details on quantification and errors, making it hard to judge the reproducibility of the data and the specificity of Gal3-LPSpa interaction (see also below). In addition, the Discussion is very much limited. How are the results compared to previous literature of similar nature? What is new in our understanding of Gal3-LPS interaction? How will the findings direct future therapeutic development? What is the implication of Gal3-LPSpa interaction on the pathology of P. aeruginosa? What are the limitations of the report? Without these details, the reviewer cannot envision the applicability of current study.

I also have the following more detailed points for the authors to address:

1.      What’s the concentration of LPSpa in Figure 1? What’s the meaning of CD measurement? It was said that titration was preformed but the details were missing;

2.      Fig 2, the DLS results of different ratio of Gal3CRD to LPSpa should be given to demonstrate a mixture of separate peaks and the final merge of Gal3CRD peak to that of LPSpa. What is the error of diameter measurements in Fig. 2 and 3?

3.      Fig. 4, again, what’s the error of measurements?

4.      Table 1, how were KD1, DH1, and DS1 determined? It’s not in the Methods section.

5.      Line 215, “E71, D77”: from Fig. 7, 71 is ASP (D) and 77 is GLU (E) – please double check;

6.      LPS catalog# from SIGMA-Aldrich should be given. What is the MW to give molar concentrations?

7.      Line 292-294, the sentence “In the interaction studies was analyzed Gal3CRD / LPSpa ratio of 10:1, in the aggregation studies were investigated the LPSpa /Gal3CRD ratios of 1:1 and 1:3” needs correction.

8.      Line 309, error “20 mM NaP”;

9.      References for the methods employed should be given. 

Comments on the Quality of English Language

See above.

Author Response

Dear Editor,

please find enclosed the revised version of our manuscript entitled “Biophysical and structural characterization of the interaction between human Galectin-3 and LPS from Pseudomonas aeruginosa” to be considered for publication in the journal “International Journal of Molecular Sciences”.

We thank the referees for the insightful comments and have done everything possible to accommodate their requests and improve the manuscript. You can find below point by point all the response to reviewers.

The main changes in the manuscript are highlighted in red in the word file

III referee

The manuscript (ijms-2856899) examined the interaction between the carbohydrate recognition domain (CRD) of human galectin-3 and LPS from Pseudomonas aeruginosa by using different methods including circular dichroism (CD), fluorescence, dynamic light scattering (DLS), zeta potential measurements, isothermal titration calorimetry (ITC) and 15N-1H TROSY NMR spectroscopy. It was concluded that these findings could help drug design targeting galectin-LPS interaction for therapy against P. aeruginosa.

Overall, my major issue with the manuscript is on data quantification. Most of the data are only described qualitatively without details on quantification and errors, making it hard to judge the reproducibility of the data and the specificity of Gal3-LPSpa interaction (see also below). In addition, the Discussion is very much limited.

Following referee observation, The details on quantifications and errors have been added in the figures modifying Figure 2, 3 and 4 accordingly.

How are the results compared to previous literature of similar nature? What is new in our understanding of Gal3-LPS interaction?

At the best of our knowledge, all the studies known in literature regarding Gal3-LPS interaction are reported in the text. There are other interaction studies between proteins and LPS such as that reported by  Brandenburg, K., Lehwark-Yvetot, G. (1999). “Characterization of the binding of endogenous proteins to endotoxins: binding stochiometry and protein secondary structure. In: Greve, J., Puppels, G.J., Otto, C. (eds) Spectroscopy of Biological Molecules: New Directions. Springer, Dordrecht. https://doi.org/10.1007/978-94-011-4479-7_6 analysing the interaction between albumin and LPS but this is far from the focus of our manuscript.

How will the findings direct future therapeutic development? What is the implication of Gal3-LPSpa interaction on the pathology of P. aeruginosa? What are the limitations of the report? Without these details, the reviewer cannot envision the applicability of current study.

We thank the reviewer for the suggestion and in order to enrich the discussion we have added some sentences in the Discussion section.

I also have the following more detailed points for the authors to address:

  1. What’s the concentration of LPSpa in Figure 1? What’s the meaning of CD measurement? It was said that titration was preformed but the details were missing;

We agree with referee and Figure 1 has been modified accordingly and the details added in the text.

  1. Fig 2, the DLS results of different ratio of Gal3CRDto LPSpa should be given to demonstrate a mixture of separate peaks and the final merge of Gal3CRD peak to that of LPSpa. What is the error of diameter measurements in Fig. 2 and 3?
  2. Fig. 4, again, what’s the error of measurements?

Regarding point 2 and 3 the details have been added in the text.

  1. Table 1, how were KD1, DH1, and DS1determined? It’s not in the Methods section.

Information was added in the MM section.

  1. Line 215, “E71, D77”: from Fig. 7, 71 is ASP (D) and 77 is GLU (E) – please double check;

Done

  1. LPS catalog# from SIGMA-Aldrich should be given. What is the MW to give molar concentrations?

Such information has been inserted in the text. MW used to give molar concentrations after treatment is 30 KDa.

  1. Line 292-294, the sentence “In the interaction studies was analyzed Gal3CRD / LPSpa ratio of 10:1, in the aggregation studies were investigated the LPSpa /Gal3CRD ratios of 1:1 and 1:3” needs correction.

      If we understand correctly what the referee wants to mean there is a reason why we choosed different ratios: the first experiment has been designed to verify the capture of the protein by the LPS and in order to observe the protein alone, we worked necessarily at high concentration (200 mM). The disaggregating effect was followed at lower LPS concentration because the modifications of the aggregation state at low concentration is more easily detectable.Thanks to the observation of the referee, we have modified the text accordingly.

  1. Line 309, error “20 mM NaP”;

We have corrected the buffer

  1. References for the methods employed should be given. 

OK

Round 2

Reviewer 2 Report

Comments and Suggestions for Authors

The authors have made suitable revisions to the paper

Author Response

The authors have made suitable revisions to the paper

Thank you very much for your positive and useful comments

Reviewer 3 Report

Comments and Suggestions for Authors

Overall I am not satisfied with the revision with issues raised previously not addressed.

1.      There are no statistics in the data presented or the statistical methods used. Is it standard deviation or standard error? Is the difference significant? At what levels?

2.      Three words of “a partial denaturation” cannot summarize the meaning of a CD measurement. The authors have to do better. Without quantitative analyses of the spectra, there would be no gain on insight of the measurement.

3.      It is said that “The variation in size in the presence of different ratios of Gal3CRD to LPSpa up to a ratio of 1:10 was investigated…”; however, the results (Fig 2) did not show the steps of change but only two extremes;

4.      “Disaggregating effect Gal3 on LPSpa could significantly lower its biological activity (line 143)” – I am not sure how the conclusion was derived. What is the relationship between the size of LPS particle and its biological activity? And what kind of biological activity are we talking about?

5.      For the ITC experiments: “The best fit is obtained adding three sequential binding molecules (line 166-167)”, “The best data fitting is obtained adding three sequential binding molecules (line 343-344).”  - the added sentences do not explain how were KD1, DH1, and DS1 determined. What are the “three sequential binding molecules”? What was the equation used?

6.      Where was the MW of L9143 of 30kDa from?

Author Response

We are sorry that the referee was not satisfied. We responded to the new points raised 

  1. There are no statistics in the data presented or the statistical methods used. Is it standard deviation or standard error? Is the difference significant? At what levels?

We agree with the referee, we have added the standard deviation  and the significance with T student’s test. Figure 4 has been modified accordingly.

  1. Three words of “a partial denaturation” cannot summarize the meaning of a CD measurement. The authors have to do better. Without quantitative analyses of the spectra, there would be no gain on insight of the measurement.

As already described in Pirone et al. IJMS 2022 (ref. 9)  the spectrum of Gal3 is not a typical spectrum of a protein with beta sheets but derives from particular characteristics of the topological arrangement, such as the length of the filaments, intra/intersheet twists or b-turns producing a spectrum with a minimum around 220 nm. It is not possible to carry out a quantitative affordable analyis of the spectra due to the noise of data dump around 200 nm in our experimental conditions. Nevertheless, we can declare that in the presence of LPS we assist to a partial denaturation corroborated by the disapperance of the positivity at around 200 nm. Such concept has been added in the text.

  1. It is said that “The variation in size in the presence of different ratios of Gal3CRD to LPSpa up to a ratio of 1:10 was investigated…”; however, the results (Fig 2) did not show the steps of change but only two extremes;

Thanks to the referee for the observation, but we focused on Figure 1 A and B, anyway in this experiment our focus is to verify the interaction between Gal3 and LPS and not a dose-response characterization. Therefore, we modified the text accordingly “up to a ratio was substituted with “The variation in size of Gal3CRD with LPSpa in a ratio of 1:10 “because a more detailed characterization was carried out by ITC experiment.

  1. “Disaggregating effect Gal3 on LPSpa could significantly lower its biological activity (line 143)” – I am not sure how the conclusion was derived. What is the relationship between the size of LPS particle and its biological activity? And what kind of biological activity are we talking about?

In addition to protecting bacteria from their environment, LPS can also act as an effector molecule by activating the host immune response against microbial pathogens (Di Grazia, A., Cappiello, F., Cohen, H. et al. D-Amino acids incorporation in the frog skin-derived peptide esculentin-1a(1-21)NH2 is beneficial for its multiple functions. Amino Acids 47, 2505-2519 (2015). https://doi.org/10.1007/s00726-015-2041-y). In fact, when released from bacteria following their cell death or division, it forms micelles, which are the active form of the endotoxin.

Thank you once again to the referee, we added an explanatory sentence in the text between the quotation marks “Data from the literature (Rosenfeld Y, Sahl HG, Shai Y (2008) Parameters involved in antimicrobial and endotoxin detoxification activities of antimicrobial peptides. Biochemistry 47:6468–6478) show that the biological activity of some antimicrobial peptides lies in their potent activity to detoxify LPS through the breakdown of LPS aggregates. The activity of Gal3 on LPS can be hypothesized in this context.”

  1. For the ITC experiments: “The best fit is obtained adding three sequential binding molecules (line 166-167)”, “The best data fitting is obtained adding three sequential binding molecules (line 343-344).” - the added sentences do not explain how were KD1, DH1, and DS1 determined. What are the “three sequential binding molecules”? What was the equation used?

I am sorry if we did not meet the referee's requests but the mathematical processing of the results is reported in “ITC Data Analysis in Origin®-Tutorial Guide Version 7.0 - January 2004-Origin® scientific plotting software to analyze calorimetric data”- as already cited in  MM section where we had indicated the software.

  1. Where was the MW of L9143 of 30kDa from?

As reported in literature, after treatment, LPS was checked via SDS-PAGE 12.5% following silver nitrate gel staining. Kazue Hatano, et al. Pseudomonas aeruginosa Lipopolysaccharide: Evidence that the 0 Side Chains and Common Antigens Are on the Same Molecule JOURNAL OF BACTERIOLOGY, 1993, P. 5117-5128. Vol. 175, No. 16.

Round 3

Reviewer 3 Report

Comments and Suggestions for Authors

To increase reproducibility, the Methods section should contain sufficient details to show how the experiments were performed and how data were analyzed. However, even after repeated requests of the reviewer, the manuscript is still not up to the standard of report without a statistical analysis section, without details of how the data are analyzed or presented. This is an issue all over the manuscript, not just one particular graph. Similarly, for the ITC experiment, the equation for curve fitting of the results using the Origin software to derive the parameters in Table 1 should be given so that it can be reproduced even without access to Origin. Molecular weight of LPS is needed for calculation of its molar concentration as stated in the manuscript although it is not a common practice given the complexity and variability of LPS. The reference supplied (Hatano et al. 1993) does not appear to contain the requested information and a proper reference should be cited to justify the MW of 30 kDa for L9143. By the way, the reference cited (#42) for the preparation of Gal3CRD is wrong and should be #8.

Author Response

Round III:

To increase reproducibility, the Methods section should contain sufficient details to show how the experiments were performed and how data were analyzed.

However, even after repeated requests of the reviewer, the manuscript is still not up to the standard of report without a statistical analysis section, without details of how the data are analyzed or presented. This is an issue all over the manuscript, not just one particular graph.

We apologize for not having sufficiently clarified some parts of the materials and methods and hope that with this additional revision we will be able to meet the demands of the reviewer. We feel that we have answered all reviewer requests. Thanks to your comments the MM section has been greatly improved and made clearer. Regarding statistical analysis section, where necessary, thanks to reviewer observation, it has been inserted (i.e. Figure 4).

Similarly, for the ITC experiment, the equation for curve fitting of the results using the Origin software to derive the parameters in Table 1 should be given so that it can be reproduced even without access to Origin.

Each ITC instrument comes with the corresponding software, so all researchers using this instrument have the methods for data processing at their disposal. In each case we have added in the materials and methods the specific name of the instrument software used in this manuscript (ITC Data Analysis in Origin: https://www.isbg.fr/IMG/pdf/itc_data_analysis_in_origin.pdf).  Because the description of the software is available online, the mathematical equations used for fitting the curves can be extrapolated from it.

Molecular weight of LPS is needed for calculation of its molar concentration as stated in the manuscript although it is not a common practice given the complexity and variability of LPS. The reference supplied (Hatano et al. 1993) does not appear to contain the requested information and a proper reference should be cited to justify the MW of 30 kDa for L9143.

Different LPS preparations in several manuscripts report different MW ranging from 4 kDa to 100 kDa depending on the treatment. Therefore, we have reported our experimental conditions and the reference how we determined the MW. We implemented MM section with more details changing the “LPS preparation” paragraph with the follow sentences: “LPS is a heterogeneous molecule and tends to form aggregates of varying sizes. However, when treated with detergents, ultrasound and heat, a population of molecules with molecular weights between 30 kDa and 100 kDa can be obtained. In our experiment, 1 mg of LPS from P. aeruginosa 10 (L9143, SIGMA-Aldrich) was resuspended in H2O steryl or in 20 mM sodium phosphate, 150mM NaCl, pH 7.4. The solution was mixed with the aid of a vortex mixer and then sonicated at 50°C for 30 minutes in the sonicator bath. After treatment, LPS MW was checked by SDS-PAGE with silver nitrate gel staining. In our experiments, solutions from 0.5 to 20 mM were used”.

By the way, the reference cited (#42) for the preparation of Gal3CRD is wrong and should be #8.

Done

Round 4

Reviewer 3 Report

Comments and Suggestions for Authors

Thanks for taking care of my comments. I have no more questions.